# Comparison of Surgical Outcomes between Single-Port Laparoscopic Surgery and Da Vinci Single-Port Robotic Surgery

**DOI:** 10.3390/jpm13020205

**Published:** 2023-01-24

**Authors:** Jeong-Min Kim, Seon-Mi Lee, Aeran Seol, Jae-Yun Song, Ki-Jin Ryu, Sanghoon Lee, Hyun-Tae Park, Hyun-Woong Cho, Kyung-Jin Min, Jin-Hwa Hong, Jae-Kwan Lee, Nak-Woo Lee

**Affiliations:** 1Department of Obstetrics and Gynecology, Korea University College of Medicine, 73 Koreadae-ro, Seongbuk-gu, Seoul 02841, Republic of Korea; 2Department of Obstetrics and Gynecology, Korea University College of Medicine, 148 Gurodong-ro, Guro-gu, Seoul 08308, Republic of Korea; 3Department of Obstetrics and Gynecology, Korea University College of Medicine, 123 Jeokgeum-ro, Danwon-gu, Ansan-si 15355, Gyeonggi-do, Republic of Korea

**Keywords:** single-port laparoscopic surgery, single-port robotic surgery, minimally invasive surgery

## Abstract

Background: The aim of this study is to compare the surgical outcomes of single-port laparoscopic surgery (SPLS) and single-port robotic surgery (SPRS). Methods: We retrospectively analyzed patients who underwent a hysterectomy, ovarian cystectomy, or myomectomy with SPLS or SPRS from January 2020 to July 2022. Statistical analyses were performed using the SPSS chi-square test and student’s *t*-test. Results: A total of 566 surgeries including single-port laparoscopic hysterectomy (SPLH; *n* = 148), single-port robotic hysterectomy (SPRH; *n* = 35), single-port laparoscopic ovarian cystectomy (SPLC; *n* = 207), single-port robotic ovarian cystectomy (SPRC; *n* = 108), single-port laparoscopic myomectomy (SPLM; *n* = 12), and single-port robotic myomectomy (SPRM; *n* = 56). The SPRH, SPRC, and SPRM groups had a shorter operation time than the SPLS group, although the results were not statistically significant (SPRH vs. SPLH, *p* = 0.134; SPRC vs. SPLC, *p* = 0.098; SPRM vs. SPLM, *p* = 0.202). Incisional hernia occurred as a postoperative complication in two patients only in the SPLH group. Postoperative Hb changes were lower in the SPRC and SPRM groups than in the SPLC and SPLM groups (SPRC vs. SPLC, *p* = 0.023; SPRM vs. SPLM, *p* = 0.010). Conclusions: Our study demonstrated that the SPRS had comparable surgical outcomes when compared to the SPLS. Therefore, the SPRS should be considered a feasible and safe option for gynecologic patients.

## 1. Introduction

Minimally invasive surgery (MIS) in gynecology requires only a small incision in the abdomen to insert a laparoscopic camera and various surgical instruments, unlike conventional open surgery with a large incision site. MIS includes laparoscopic, robotic, and natural orifice transluminal surgeries [1,2]. From 2004 to 2007, the number of MIS done in the United States has increased with the trend of increasing interest regarding MIS. In particular, the rate of robotic surgeries performed in 2012 was significantly increased by 50% compared to 27.9% of robotic surgeries performed in 2007 [1]. According to a retrospective review conducted by Bingmer et al., laparotomy decreased by 34.9% from 135.5 to 88.2 from 2000 to 2018, but laparoscopic surgery showed a dramatic increase of 462% from 23.6 to 135.6 [3]. MIS has been widely performed due to the reduction in postoperative pain, reduced hospital stay, rapid recovery period, and small postoperative wounds [4,5].

Although single-site surgery has cosmetic benefits, single-port laparoscopic surgery (SPLS) has surgical limitations in angulation and device manipulation. Moreover, the training period for SPLS is longer than for robotic surgery or open surgery [4,6,7]. On the other hand, robotic surgery allows secured visibility even in a limited space as the camera can be moved to a desired location in the abdominal cavity. Moreover, it prevents hand tremors, provides three-dimensional (3D) high-definition images, uses a surgical instrument with the EndoWrist, and allows the operator to perform sophisticated surgery in a less tiring state. This applies equally to single-port robotic surgery (SPRS) [4,6]. However, the disadvantages of robotic surgery include more expensive costs than laparoscopic surgery, and the robotic surgery must be performed in a fixed patient position after robot docking and instrument insertion are completed. In addition, the instruments used in SPRS are limited compared with SPLS or multi-port robotic surgery [4].

In many studies comparing the surgical outcomes of SPLS, multi-port laparoscopic surgery (MPLS), SPRS, and multi-port robotic surgery (MPRS), there were no statistical differences in operation time, postoperative hemoglobin (Hb) changes, hospital stay, and additional pain medication [2,8,9,10]. However, there are few studies that have compared the outcomes of SPLS and SPRS. Therefore, this study aims to compare the surgical outcomes of SPLS and SPRS.

## 2. Materials and Methods

### 2.1. Patient Selection and Data Collection

We retrospectively evaluated data from 566 women who underwent a hysterectomy, ovarian cystectomy, and myomectomy using SPLS or SPRS at the Korea University Anam Hospital from January 2020 to July 2022. Finally, 148 patients underwent single-port laparoscopic hysterectomy (SPLH), 35 patients underwent single-port robotic hysterectomy (SPRH), 207 patients underwent single-port laparoscopic ovarian cystectomy (SPLC), 108 patients underwent single-port robotic ovarian cystectomy (SPRC), 12 patients underwent single-port laparoscopic myomectomy (SPLM), and 56 patients underwent single-port robotic myomectomy (SPRM).

All procedures were performed by six gynecological surgeons at the Korea University Anam Hospital. There may be differences in surgical proficiency and surgical experience of the six surgeons, and the SP entry system and energy device used during the operation may be different depending on the surgeon’s preference. However, the overall surgical procedures for hysterectomy, ovarian cystectomy, and myomectomy using a single port is not very different for all six surgeons. In SPLS, approximately a 1.5-2.0 cm umbilical vertical incision was made with the Hasson method. One of the SP entry systems including LAPSINGLE (SEJONG Medical Co., Paju, Korea), ONE-PORT plus (MEDFINE CO., Hanam, Korea), and Glove Port (NELIS CO., Bucheon, Korea) was used for instrument application. Hysterectomy was defined as completely removing the patient’s uterus and suturing the vaginal stump. Ovarian cystectomy was defined as peeling and the removal of the cystic tumor lesion from the ovarian tissue and suturing residual normal ovarian tissue. Myomectomy was defined as the removal of fibroids from the uterus and suturing of the myometrium layer and serosal layer of the uterus. After surgery, such as a hysterectomy, myomectomy, or ovarian cystectomy, the umbilical incision was closed layer-by-layer from the fascia to the skin.

A da Vinci SP surgical system (Intuitive Surgical Inc., Sunnyvale, CA, USA) was used for SPRS including SPRH, SPRC, and SPRM. In SPRS, approximately a 2.5 cm umbilical vertical incision was made with the Hasson method. One of the SP entry systems, such as Lapsingle Vision SP (SEJONG Medical Co., Paju, Korea), UNI-PORT SP (DALIM Medical Co., Bucheon, Korea), and Glove Port SP (NELIS Co., Bucheon, Korea), was used for instrument application. The SP entry system used in SPRS has space for a 25 mm da Vinci SP cannula that was different from the SP site instruments used in SPLS. The surgical procedures including SPRH, SPRC, and SPRM were not different from those of SPLH, SPLC, and SPLM.

We retrieved the hospital medical records of the study subjects. Surgical time was defined as the time from the first skin incision to the final skin closure, which included docking time for SPRS. The robot-docking time was defined as the time required to place the robotic instruments on the port placement inserted into the patient after the robot patient cart was driven. The robot-console time was defined as the time it took for the surgeon to sit on the robot console and perform the surgery. Robot-docking time and robot-console time were included in the surgical outcomes of SPRS groups. Postoperative Hb change-value was obtained by comparing the difference between Hb reported in complete blood count on postoperative day 1 and preoperative Hb.

### 2.2. Statistical Analysis

A student’s *t*-test was conducted to compare and analyze continuous variables between the SPLS and SPRS groups, and categorical variables were analyzed using a chi-square or Fisher’s exact test. Statistical significance was defined as a *p*-value < 0.05. All analyses were performed using SPSS Statistics for Windows (version 25.0; SPSS Inc., Chicago, IL, USA).

### 2.3. Ethics

The study protocol and waiver of informed consent were approved by the Institutional Review Board (IRB) of the Korea University Anam Hospital (IRB number: 2022AN0556). All procedures were performed in accordance with the relevant guidelines and regulations of the institution.

## 3. Results

The patient demographics are shown in Table 1. There was no significant statistical difference with respect to age, body mass index (BMI), and parity in SPLH vs. SPRH, SPLC vs. SPRC, and SPLM vs. SPRM. In the SPLH and SPRH groups, the four most common indications of operation were myoma (SPLH, 64.2%; SPRH, 62.9%), adenomyosis (SPLH, 14.2%; SPRH, 17.1%), cervical dysplasia (SPLH, 14.2%; SPRH, 14.3%), and endometrial lesion (SPLH, 7.4%; SPRH, 2.9%). In the SPLC and SPRC groups, there was no significant difference in terms of bilateral ovarian cyst (SPLC, 25.1%; SPRC, 16.7%; *p* = 0.087), and largest ovarian cyst (SPLC, 5.78 ± 2.35 cm; SPRC, 6.70 ± 4.70 cm; *p* = 0.058). However, the rate of previous abdominal surgery history was higher in the SPRC group than in the SPLC group (SPLC, 9.2%; SPRC, 23.1%; *p* = 0.001). In the SPLM and SPRM groups, there was no significant statistical difference with respect to the number of myomas (SPLM, 2.42 ± 2.31; SPRM, 2.20 ± 1.77; *p* = 0.713) and the size of the largest myoma (SPLM, 6.57 ± 3.32 cm; SPRM, 6.06 ± 2.83 cm; *p* = 0.586). The two most common locations of the largest uterine myoma in SPLM and SPRM were anterior (SPLM, 33.3%; SPRM, 39.3%) and posterior (SPLM, 33.3%; SPRM, 37.5%).

The surgical outcomes are presented in Table 2. Variables such as surgical time, hospital stay, adhesiolysis, and postoperative transfusion were not different between SPLH vs. SPRH, SPLC vs. SPRC, and SPLM vs. SPRM. In the SPLH and SPRH groups, there was no significant difference regarding uterine weight (SPLH, 362.94 ± 534.20 g; SPRH, 308.31 ± 231.03 g; *p* = 0.555). However, the pathology result was different between the SPLC and SPRC groups (*p* < 0.001). The two most common histologic results in the SPLC group were teratoma (54.6%) and endometriosis (45.4%), and in the SPRC group were cystadenoma (38.9%) and others such as abscess or inclusion cyst (43.5%). In the SPLM and SPRM groups, the most common type of myoma was the intramural type, but there was no significant statistical difference (SPLM, 75.0%; SPRM, 76.8%, *p* = 0.802).

## 4. Discussion

Surgical outcomes including surgical time, postoperative Hb change, hospital stay, operative complications, and postoperative transfusion were analyzed. In this study, there was a statistically significant decrease in postoperative Hb in the SPRC and SPRM groups compared with the SPLC and SPLM groups. Although not statistically significant, the surgical times were shorter in the SPRS group than in the SPLS group. Except for an incisional hernia that occurred in two cases of the SPLH group, there was no other operative complication and no conversion to laparotomy.

Single-site surgery has a cosmetic advantage because it leaves a single scar on the umbilicus. However, there are disadvantages, such as limited space for surgery and collisions between surgical instruments. Since the laparoscopic hysterectomy using a single umbilical puncture performed by Pelosi in 1991 [10], surgical instruments, surgical systems, and surgical techniques have been developed to overcome the limitation of single-incision approaching surgery [8,11].

Two studies have compared the surgical outcomes of SPRH and multi-port robotic hysterectomy (MPRH). There were no statistically significant differences with respect to the characteristics of participants, surgical time, and hospital stay in both the study conducted by Park et al. [2] and the study conducted by Bogliolo et al. [12]. However, in a retrospective study of 104 women conducted by Bogliolo et al., the EBL in the SPRH group was lower than the MPRH group (SPRH, 46 ± 52 mL; MPRH, 150 ± 151 mL, *p* < 0.001) [12]. The operation time of the SPRH group in our study was 114 ± 71 min, which was shorter than 165 min and 144 min in the two studies mentioned above [2,12]. In this study, the average uterine weight after the SPRH group was approximately 308.31 g, which was less than the average uterine weight of 445.9 g reported by Park et al. [2] and greater than the average uterine weight of approximately 137.69 g reported by Bogliolo et al. [12]. In another study comparing surgical outcomes between SPLH and multi-port laparoscopic hysterectomy (MPLH) groups, the surgical outcomes, such as surgical time, EBL, postoperative Hb change, the size of the uterus, and hospital stay were similar for the SPLH and MPLH groups [13]. However, the pain scores at 24 and 36 h after the operation were lower in the SPLH group than in the MPLH group, which were statistically significant (pain score at 24 h in the SPLH group, 2.5 ± 0.7; pain score at 24 h in the MPLH group, 3.5 ± 0.8; *p* = 0.01 vs. pain score at 36 h in the SPLH group, 1.7 ± 1.2; pain score at 36 h in the MPLH group; 2.9 ± 1.1, *p* = 0.01) [13]. In this study, the surgical time of the SPLH group was 128.69 ± 50.46 min, which was longer than the surgical time of the SPLH group (119 ± 32 min) reported by Kim et al., Unlike this study, which measured uterine weight, the study conducted by Kim et al., compared based on the diameter of the uterus. Therefore, it was difficult to evaluate the difference from surgical time. In the aforementioned studies, there was no operative complication such as adjacent organ damage, conversion to laparotomy, and reoperation. However, our study showed two incisional hernias in two patients who underwent SPLH. The umbilical hernia is divided into early-onset hernia, which occurs immediately after surgery, and late-onset hernia, which occurs several months after surgery [14]. In one cohort study, the umbilical hernia incidence is higher in obese patients than in normal BMI patients [15]. On the other hand, the other study reported that the umbilical hernia occurred in approximately two-thirds of all patients underwent single-site surgery, regardless of the patient’s BMI. Thus, close monitoring is necessary after single-site surgery [14]. There was no statistically significant difference for patient characteristics including BMI in two patients in the SPLH group compared to the SPLH group. Further longitudinal research is needed to evaluate the occurrence of umbilical hernia after long-term follow-up in patients who underwent single-site surgery and to determine the factors that significantly increase the risk of umbilical hernia.

In a retrospective study of 25 patients who underwent SPLC, the surgical time (67.2 min) was shorter than the surgical time (92.10 ± 55.06 min) in our SPLC group [16]. Because the retrospective study did not present the size of the ovarian cyst, we did not compare the size of the ovarian cyst between the retrospective study and this study. Unlike this study, the retrospective study added a 3 mm trocar inserted in the left iliac area throughout a Z-shape incision in addition to the umbilical incision. The use of an additional trocar can help overcome collisions among laparoscopic instruments. It is not surprising that the SPLH group in this study, which did not use an additional trocar, had a longer surgical time considering the reason mentioned above. Yun et al., analyzed the surgical outcomes of SPRC, the surgical time (140.54 ± 62.37 min) in Yun’s study was longer than the SPRC group (81.90 ± 45.07 min) in our study. In addition, the postoperative Hb change (1.88 ± 0.74 g/dL) in Yun’s study was higher than the SPRC’s postoperative Hb change (1.59 ± 1.08 g/dL) in this study [17]. In Yun’s study, the average size of an ovarian cysts was 9.63 ± 3.32 cm and the rate of adhesiolysis was 56.3%. In our study, the size of an ovarian cysts was 6.70 ± 4.50 cm and adhesiolysis was done in 19.4% of the cases. Our study had smaller ovarian cysts and performed less adhesiolysis procedures, and these reasons may have influenced the shorter surgical time of the SPRC group in this study.

Lee et al., analyzed surgical outcomes in patients who underwent SPRM [18]. In a study conducted by Lee et al., the surgical time (149.9 ± 72.9 min) was longer than the surgical time (134.55 ± 63.39 min) of the SPRM group in our study. When comparing our study with the prospective study by Lee et al., there were the following differences. The largest size of the uterine myoma (7.6 ± 2.9 cm) was larger than the largest diameter of the uterine myoma in this study (6.06 ± 2.83 cm). In a prospective randomized trial comparing surgical outcomes between SPLM and multi-port laparoscopic myomectomy (MPLM), there was no statistically significant difference in surgical outcomes, such as surgical time, hospital stay, and postoperative pain between the two groups. However, the patients’ satisfaction with postoperative wounds was higher in the SPLM group than in the MPLM group [19]. Comparing the SPLM group of the above-mentioned study and the SPLM group in this study, although the largest size of myoma (6.57 ± 3.32 cm) in this study was smaller, the surgical time (160.42 ± 61.62 min) was longer than the largest size of myoma (6.7 ± 1.9 cm) and the surgical time (109.3 ± 49.6 min) in the above-mentioned study. In the above-mentioned prospective study, only one or two layers were performed at a ratio of approximately 1:1 when the residual uterine tissue was sutured after a myomectomy. However, in this study, the suturing process was more advanced in a way that the myometrium suture was performed over the basic two layers, and the baseball suture was performed between the rest of the uterine serosa and myometrium. The differences in the surgical procedures caused the difference in the operation times.

As a result of this paper, there was no statistical difference, but the surgical time of the SPRC group was shorter than that of the SPLC group. Although the rate of the previous abdominal surgery history was higher in the SPRC group than in the SPLC group, the rate of bilateral ovarian cystectomy was higher in the SPLC group and the rate of adhesiolysis was higher in the SPLC group than in the SPRC group. These factors may have affected the outcome of shorter surgical time in the SPRC group than in the SPLC group. Most previously published papers compared the surgical outcomes of patients who underwent single-site surgery and multi-site surgery throughout laparoscopic surgery or robotic surgery. However, few studies have compared the outcomes between SPLS and SPRS. In a study comparing the surgical outcomes of SPLC and SPRC conducted by Kim et al. in 2020, the surgical outcomes of the SPRC group were comparable to those of the SPLC group. In addition, Kim et al., reported that the learning curve was faster in the SPRC group than the SPLC group [20].

This study has several limitations. First, heterogeneity could be reflected because the outcomes of surgery performed by several surgeons were compared and analyzed for each group. Second, as the study was conducted at a single institution, the number of study participants was small compared to studies conducted at multiple institutions. Third, there was a limitation in the longitudinal follow-up of each patient in the retrospective study. Fourth, the selection bias might be reflected in the process of selecting patient information for study participants because the information was obtained retrospectively from patient medical records.

Nevertheless, the strength of this study was that it is one of the few studies that has compared the surgical outcomes of hysterectomy, ovarian cystectomy, and myomectomy performed using SPLS or SPRS.

## 5. Conclusions

The surgical outcomes of SPRS including the SPRH, SPRC, and SPRM groups, were comparable to those of SPLS, such as the SPLH, SPLC, and SPLM groups. Especially, postoperative Hb change was significantly decreased in the SPRC and SPRM groups compared to the SPLC and SPLM groups. There was no operative complication except for two cases in the SPLH group. Therefore, this study is meaningful in that it suggests that SPRS can be proposed as a relatively safe and feasible surgical method for patients who desire to undergo single-site surgery.

## Figures and Tables

**Table 1 jpm-13-00205-t001:** The patients’ characteristics of SPLH, SPRH, SPLC, SPRC, SPLM, and SPRM groups.

Variables	SPLH (*n* = 148)	SPRH (*n* = 35)	*p*-Value	SPLC (*n* = 207)	SPRC (*n* = 108)	*p*-Value	SPLM (*n* = 12)	SPRM (*n* = 56)	*p*-Value
Age (years)	50.89 ± 8.26	49.71 ± 7.86	0.448	28.78 ± 7.44	31.79 ± 10.31	0.011	41.42 ± 7.85	36.98 ± 7.14	0.059
BMI (kg/m^2^)	24.64 ± 3.86	24.40 ± 3.74	0.746	22.07 ± 3.69	22.86 ± 4.39	0.095	23.92 ± 5.02	23.09 ± 4.19	0.550
Parity	1 (0–6)	1 (0–4)	0.982	0 (0–3)	0 (0–3)	0.034	0 (0–2)	0 (0–3)	0.510
Previous abdominal									
surgery history			0.171			0.001			0.365
No	94 (63.9%)	18 (51.4%)		188 (90.8%)	83 (76.9%)		9 (75.0%)	49 (87.5%)	
Yes	53 (36.1%)	17 (48.6%)		19 (9.2%)	25 (23.1%)		3 (25.0%)	7 (12.5%)	
Comorbidity			0.888			0.151			<0.001
None	106 (71.6%)	24 (68.6%)		197 (95.2%)	104 (96.3%)		0 (0.0%)	56 (100.0%)	
Cardiovascular	23 (15.9%)	6 (17.1%)		3 (1.4%)	2 (1.9%)		4 (33.3%)	0 (0.0%)	
DM	7 (4.7%)	1 (2.9%)		1 (0.5%)	2 (1.9%)		0 (0.0%)	0 (0.0%)	
Others	11 (7.4%)	4 (11.4%)		6 (2.9%)	0 (0.0%)		8 (66.7%)	0 (0.0%)	
Preoperative Hb	12.26 ± 1.68	12.32 ± 1.44	0.864	12.97 ± 1.00	12.50 ± 1.26	0.001	13.58 ± 0.64	12.65 ± 1.43	0.033

Note: Data are shown as the M ± SD or N (%). SPLH, single-port laparoscopic hysterectomy; SPRH, single-port robotic hysterectomy; SPLC, single-port laparoscopic ovarian cystectomy; SPRC, single-port robotic ovarian cystectomy; SPLM, single-port laparoscopic myomectomy; SPRM, single-port robotic myomectomy; BMI, body mass index; Hb, hemoglobin; DM, diabetes mellitus.

**Table 2 jpm-13-00205-t002:** Surgical outcomes of SPLH, SPRH, SPLC, SPRC, SPLM, and SPRM.

	SPLH (*n* = 148)	SPRH (*n* = 35)	*p*-Value	SPLC (*n* = 207)	SPRC (*n* = 108)	*p*-Value	SPLM (*n* = 12)	SPRM (*n* = 56)	*p*-Value
Surgical time (minutes)	128.69 ± 50.49	114.71 ± 44.20	0.134	92.10 ± 55.06	81.90 ± 45.07	0.098	160.42 ± 61.62	134.55 ± 63.39	0.202
Docking time (minutes)	-	3.66 ± 1.37	N/A	-	3.59 ± 2.08	N/A	-	3.48 ± 1.98	N/A
Postperative Hb change (g/Dl)	1.5 ± 1.13	1.53 ± 1.04	0.892	1.88 ± 1.03	1.59 ± 1.08	0.023	2.85 ± 1.62	1.75 ± 1.20	0.010
Hospital stay (days)	4.55 ± 1.26	4.54 ± 1.01	0.985	4.37 ± 0.96	4.55 ± 1.86	0.262	5.17 ± 2.65	4.57 ± 1.12	0.461
Intraoperative complication									
No	148 (100.0%)	35 (100.0%)		207 (100.0%)	108 (100.0%)		12 (100.0%)	56 (100.0%)	
Yes	0 (0.0%)	0 (0.0%)		0 (0.0%)	0 (0.0%)		0 (0.0%)	0 (0.0%)	
Postoperative complication			1.000						
No	146 (98.6%)	35 (100.0%)		207 (0.0%)	108 (100.0%)		12 (100.0%)	56 (100.0%)	
Yes	2 (1.4%)	0 (0.0%)		0 (0.0%)	0 (0.0%)		0 (0.0%)	0 (0.0%)	
Conversion to laparotomy									
No	148 (100.0%)	35 (100.0%)		207 (100.0%)	108 (100.0%)		12 (100.0%)	56 (100.0%)	
Yes	0 (0.0%)	0 (0.0%)		0 (0.0%)	0 (0.0%)		0 (0.0%)	0 (0.0%)	
Adhesiolysis									
			0.084			0.136			0.136
No	121 (81.8%)	24 (68.4%)		151 (72.9%)	87 (80.6%)		9 (91.7%)	42 (94.6%)	
Yes	27 (18.2%)	11 (31.6%)		56 (27.1%)	21 (19.4%)		3 (27.1%)	4 (19.4%)	
Postoperative transfusion			1.000			1.000			1.000
No	146 (98.6%)	35 (100.0%)		206 (99.5%)	107 (99.1%)		11 (99.5%)	55 (99.1%)	
Yes	2 (1.4%)	0 (0.0%)		1 (0.5%)	1 (0.9%)		1 (0.5%)	1 (0.9%)	

Note: Data are shown as the M ± SD or N (%). SPLH, single-port laparoscopic hysterectomy; SPRH, single-port robotic hysterectomy; SPLC, single-port laparoscopic ovarian cystectomy; SPRC, single-port robotic ovarian cystectomy; SPLM, single-port laparoscopic myomectomy; SPRM, single-port robotic myomectomy; N/A, not available.

## Data Availability

Data sharing is not applicable to this study because of privacy or ethical restrictions.

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
