# Peer review of "Comparison of Surgical Outcomes between Single-Port Laparoscopic Surgery and Da Vinci Single-Port Robotic Surgery"

_jpm, 2023, doi:10.3390/jpm13020205_

Round 1

Reviewer 1 Report

The article is an interesting one, since nowadays there are many studies trying to assess whether Robotic Surgery is worth the price or we should stick to laparoscopic surgery. A debate that reminds of past times where we were debating the same thing between laparoscopic surgery and open surgery.

You said that all the surgeries were performed by six different physicians, but there was no description about their surgical prowess. Robotic surgery, as you stated, has a less steeper curve than single port laparoscopic surgery, thereby it could be important to compare required experience to perform SPLS vs SPRS since there has been no difference found in Surgical Time.

In the statistical analysis you performed a Student's T test, but there is no indication if you analysed the sample to check for normality  with a Kolmogorov-Smirnov or a Shapiro-Wilk test. It is important to make it clear, since sometimes you could deduce that the distribution of your sample has no importance in the tests you can perform.

Finally, you found a statistically significant difference between SPLC y SPRC in previous abdominal surgery. Did you analise this difference? Did you discard a possible bias where the most complicated cases go for SPRS and thereby should take longer surgical time?

Author Response

Manuscript;

Comparison of Surgical Outcomes Between Single-Port Laparoscopic Surgery and Da Vinci Single-Port Robotic Surgery (Manuscript ID: jpm-2162762)

Response to Reviewer 1 Comments

Dear reviewers

Thank you for giving us the opportunity to submit a revised draft of the manuscript “ Comparison of Surgical Outcomes Between Single-Port Laparoscopic Surgery and Da Vinci Single-Port Robotic Surgery ” for publication in the JPM. We appreciate the time and effort that you and the reviewers dedicated to providing feedback on our manuscript and are grateful for the insightful comments on and valuable improvements to our paper.

We have incorporated most of the suggestions made by the reviewers. Any revisions to the manuscript be marked up using the track changes function at MS Word. Please see below, for a point-by-point response to the reviewers’ comments and concerns.

Point 1: You said that all the surgeries were performed by six different physicians, but there was no description about their surgical prowess. Robotic surgery, as you stated, has a less steeper curve than single port laparoscopic surgery, thereby it could be important to compare required experience to perform SPLS vs SPRS since there has been no difference found in Surgical Time

Response 1: Thank you for pointing this out. As you pointed out, there is bound to be a difference in surgical proficiency and surgical experience of the six surgeons. Therefore, when referring to limitaion in the discussion part of this paper, it was mentioned that heterogeneity can be reflected as a paper analyzing outcomes of operations peformed by six surgeons. In addition, in line 79 to 84 of the materials and methods part of this paper, the following phrases were added. Although each of the six surgeons has different surgical proficiency and experience, the overall surgical procedures do not differ significantly, and if there is a difference, the type of SP entry system or energy device used depends on the operator’s preference.

Point 2: In the statistical analysis you performed a Student's T test, but there is no indication if you analysed the sample to check for normality with a Kolmogorov-Smirnov or a Shapiro-Wilk test. It is important to make it clear, since sometimes you could deduce that the distribution of your sample has no importance in the tests you can perform.. 

Response 2: Thank you for pointing this out. In this study, SPLH, SPRH, SPLC, SPRC, and SPRM groups are all groups that contain more than 30 subjects. If the number of samples to be comared is 30 or more, the student’s t test can be used as an independent sample analysis method with the assumption that it follows a normal distribution. The SPRM group, which includes fewer than 30 subjects, conducted a Kolmogorov-Smirnov or a Shapiro-Wilk test before statistical analysis to check the presence of normal distribution, and then used the student’s t test to compare the average of each group.

Point 3: Finally, you found a statistically significant difference between SPLC y SPRC in previous abdominal surgery. Did you analise this difference? Did you discard a possible bias where the most complicated cases go for SPRS and thereby should take longer surgical time?

Response 3: Thank you for pointing this out. As you pointed out, the degree of history of previous abdominal operation may affect the time of operation, However, the previous history of abdominal operation is also important, but the degree of adhesion in the surgical field and how many ovarian cystectomy were performed at the time of surgery are also important factors in determining the surgical time. Therefore, as as result of this study, we added an opinion on why SPRC group had higher history of previous abdominal operation than SPLC group, but SPRC group had a shorter surgical time than SPLC group, which is presented in lines 258 to 263 of this paper.  

Reviewer 2 Report

This article compares the surgical outcomes of SPLS and SPRS. The paper is generally innovative but gives some suggestions for clinical surgical options. There are some recommendations.

line 41-44: The authors cite data from 2012 as well as 2007, which is a decade back. The authors could have added data from recent years.

line 44-58: The authors quote a lot from the same article. Can the authors provide more evidence?

line 114-134: The authors should double-check that the data in the text are consistent with the table.

Streamline the discussion section.

Author Response

Manuscript;

Comparison of Surgical Outcomes Between Single-Port Laparoscopic Surgery and Da Vinci Single-Port Robotic Surgery (Manuscript ID: jpm-2162762)

Response to Reviewer 2 Comments

Dear reviewers

Thank you for giving us the opportunity to submit a revised draft of the manuscript “ Comparison of Surgical Outcomes Between Single-Port Laparoscopic Surgery and Da Vinci Single-Port Robotic Surgery ” for publication in the JPM. We appreciate the time and effort that you and the reviewers dedicated to providing feedback on our manuscript and are grateful for the insightful comments on and valuable improvements to our paper.

We have incorporated most of the suggestions made by the reviewers. Any revisions to the manuscript be marked up using the track changes function at MS Word. Please see below, for a point-by-point response to the reviewers’ comments and concerns.

Point 1: line 41-44: The authors cite data from 2012 as well as 2007, which is a decade back. The authors could have added data from recent years.

Response 1: Thank you for pointing this out. As you mentioned, I think it is reasonable to add that the use of MIS is increasing based on recent data. So we added the lastest data-driven content to the 44th to 47th lines of this paper.  

Point 2: line 44-58: The authors quote a lot from the same article. Can the authors provide more evidence?

Response 2: Thank you for pointing this out. As you pointed out, we thought it appropriate to cite content based onmore diverse papers, se we conducted additional paper citations and reflected them in the references. Additional paper citations were carried out in the contents of lines 47 to 53 of the manuscript.

Point 3: line 114-134: The authors should double-check that the data in the text are consistent with the table.

Response 3: Thank you for pointing this out. As you pointed out, we checked again to see if the data of this study matched the data presented in the manuscript and confirmed that there was no problem. Although the results of the contents of line 114 to 134 of the manuscript were not presented in the table, it was confirmed that they were written in the manuscript consistent with the results of the analysis data conducted in this study.

Round 2

Reviewer 2 Report

Thank you for your careful reply. I think your research is valuable.